# Effects of Amino Acids Supplementation on Lipid and Glucose Metabolism in HepG2 Cells

**DOI:** 10.3390/nu14153050

**Published:** 2022-07-25

**Authors:** Shuang Wang, Soohan Jung, Kwang Suk Ko

**Affiliations:** 1Department of Nutritional Science and Food Management, Ewha Womans University, Seoul 03760, Korea; hw004027@naver.com; 2Department of Biomedical Sciences, Korea University, Seoul 02841, Korea; tngks1231@gmail.com; 3Graduate Program in System Health Science and Engineering, Ewha Womans University, Seoul 03760, Korea

**Keywords:** sulfur containing amino acids, branched chain amino acids, metabolic syndrome

## Abstract

Non-alcoholic fatty liver disease and type 2 diabetes are representing symptoms of metabolic syndrome, which is often accompanied with hepatic fat accumulation and insulin resistance. Since liver is the major site of glucose and lipid metabolism, this study aimed to understand the effects of SCAAs and BCAAs supplementations on glucose and lipid metabolism in HepG2 cells. These cells were pretreated with SAMe, betaine, taurine, and BCAA for 24 h, followed by treatments of a high concentration of glucose (50 mM) or palmitic acid (PA, 0.5 mM) for 48 h to simulate high-glucose and high-fat environments. Pretreatment of BCAA and SCAAs inhibited the fat accumulation. At the transcriptional level, glucose and PA treatment led to significant increase of mRNA gluconeogenic enzyme. The mRNA expression level of GLUT2 was decreased by 20% in the SAMe-treated group and inhibited glucose synthesis by reducing the level of gluconeogenic enzyme. After SAMe or BCAA pretreatment, the mRNA expression of lipogenic enzymes was decreased. The PPAR-γ expression was increased after BCAA pretreatment, but SAMe not only downregulated the expression of PPAR-γ, but also inhibited the expression of ChREBP approximately 20% and SREBP-1c decreased by about 15%. Taken together, the effect of SAMe on glucose and lipid metabolism is significant especially on inhibiting hepatic lipogenesis and gluconeogenesis under the metabolic syndrome environment.

## 1. Introduction

Metabolic syndrome (MetS) is determined by a series of interrelated biochemical, physiological, clinical, and metabolic factors, reflecting excess nutrition, sedentary lifestyle and resulting obesity [1,2]. The incidence of MetS is increasing not only in developed countries, but also in developing countries. MetS can be defined as any three of the following five conditions: elevated fasting glucose or type 2 diabetes, hypertension, hypertriglyceridemia, lowers high density lipoprotein (HDL) cholesterol or increased waist circumference [3]. The pathogenesis of nonalcoholic fatty liver disease (NAFLD) begins with liver fat accumulation and associated with insulin resistance, but not with other known causes of steatosis (e.g., alcohol, viruses, and drugs). In the liver, excess sugars are converted to triglycerides via de novo lipogenesis, which is increased in NAFLD [4]. Because the liver plays a central role in regulating glucose and fat metabolism, regulating the glucose and lipid metabolism, inhibiting the lipid accumulation in the liver, and promoting the utilization of glucose may be an effective means to prevent MetS [5].

The liver plays an important role in the metabolism of sulfur-containing amino acids. Methionine is an essential sulfur-containing amino acid, and approximately half of dietary methionine is metabolized in the liver. The first step of methionine metabolism is the synthesis of S-adenosylmethionine (SAMe) catalyzed by methionine adenosyltransferase (MAT) [6]. S-adenosylmethionine is found in all mammalian cells and widely distributed in the human body. It plays an important role in a series of biochemical reactions, such as enzymatic transsulfuration, transmethylation, and polyamine synthesis [7]. Betaine (N,N,N-trimethylglycine) is a soluble alkaloid that is involved in methionine recycling and phosphatidylcholine synthesis, and it is the only methyl donor that can replace folate or SAMe in humans [8]. Several studies have shown that mice fed a high-fat diet for several weeks develop NAFLD, but supplementation with betaine at the same time increased the SAMe level, prevented betaine deficiency and liver steatosis, and restored insulin sensitivity [9,10]. In particular, Asma et al. demonstrated that the supplementation of betaine can improve glucose homeostasis and reduce lipid accumulation in the liver [11]. Taurine (2-amino ethanesulfonic acid) (Tau) is one of the most abundant amino acids in the body and is found in many tissues, including the liver. Many studies have shown that taurine is involved in a variety of physiological processes such as antioxidant and Ca2^+^ transport regulation, osmoregulation, and neuromodulation [12]. Many studies have shown that taurine can reduce oxidative stress and improve alcohol-induced liver injury by inhibiting the production of reactive oxygen species (ROS) [13]. Moreover, several studies have shown that taurine has high potential in the prevention of diabetes [14,15]. Lampson et al. reported that taurine plays an insulin-like role in promoting glycolysis [16]. According to the 2010 WHO-CARDIAC survey, high urinary taurine levels are associated with lower body mass index (BMI), blood pressure, cholesterol, and sclerotic index [17].

Leucine (Leu), valine (Val), and isoleucine (Ile) are called branched-chain amino acids (BCAAs), which are essential amino acids that cannot be synthesized de novo in the human body. They are well-known as fitness supplements for muscle synthesis. They play important roles in metabolic balance and cellular functions, including energy balance and protein and lipid metabolism regulation [18]. Several studies have shown that supplementation with BCAAs increases glucose metabolism in skeletal muscle, adipose tissue, and the liver [19,20].

As mentioned above, SCAA seems to be effective for MetS related diseases such as NAFLD and type 2 diabetes, but its effect on metabolic disorders has not been fully elucidated. Therefore, in this study, the effect of SCAA on glucose and lipid metabolism was studied. In addition, the effect of BACC on the corresponding metabolic process was studied, and the regulatory mechanism of SCAA and BCAA in glucose and lipid metabolism was also studied.

## 2. Materials and Methods

### 2.1. Cell Culture

The human-hepatoma-derived cell line HepG2 (ATCC, Manassas, VA, USA) was cultured in Minimum Essential Medium (MEM) with Earle’s Balanced Salts (MEM/EBSS) (Hyclone, South Logan, UT, USA) supplemented with 10% fetal bovine serum (FBS) (Corning, VA, USA) and 1% penicillin and streptomycin (Gibco, MA, USA). Cells were incubated at 37 °C in a 5% CO_2_ humidified incubator.

### 2.2. Amino Acids Treatments

HepG2 cells (0.2 × 106 cells/well) were seeded in 6-well plates (Corning, Manassas, VA, USA) at 37 °C in a 5% CO_2_ humidified incubator for 6 h. After 6 h, cells were pretreated with 0.5 mM SAMe, 10 mM taurine, 1 mM betaine, and 4 mM BCAA mixture (leucine/isoleucine/valine = 2:1:1.2 by weight) for 24 h. S-adenosyl methionine, in the the form of S-adenosl-L-methionine disulfate P-toluenculfonate (Samoh pharm Co., LTD, Seoul, Korea), was dissolved in 0.2 M Tris-HCl Buffer (pH = 7.4) (LPS Solution, Deagu, Korea). Taurine (Sigma Aldrich, St. Louis, MO, USA) and betaine (Sigma Aldrich, MO, USA) were dissolved in phosphate buffered saline (PBS) (Hyclone, South Logan, UT, USA).

### 2.3. Glucose and Lipid Treatments

After pretreatment, the cells were grown in medium, including 50 mM D-glucose (Sigma Aldrich, St. Louis, MO, USA) and/or 0.5 mM palmitic acid (Sigma Aldrich), for 48 h. Glucose solution of 1 M was prepared by dissolving glucose in PBS. Then an appropriate amount of glucose solution was added to the prewarmed (37 °C) medium. The final concentration of glucose in medium was 50 mM. The glucose medium was filtered with 0.2 M sterile filter (Sartorius, Goettingen, Germany) before treatment. The medium containing fatty acid was prepared by combining palmitic acid (PA) with bovine serum albumin (BSA) (Sigma Aldrich). BSA was dissolved in PBS to make a 20% (*w*/*v*) BSA solution. Then 0.0056 g palmitic acid (Sigma Aldrich) was dissolved in 1 mL double-distilled water containing 100 μL 1N NaOH to make fresh 20 mM PA solution each time by incubating at 70 °C in a water bath for 15~30 min. Palmitic acid was immediately added to prewarmed (37°C) 20% BSA solution in the required amount, and then the mixture was added to prewarmed (37°C) medium. The final concentration of PA in medium was 0.5 mM, and the molar ratio of PA to BSA was 2:1. The medium containing PA was filtered to treat the cells.

### 2.4. Cell Viability Assays

Cell viability was measured by using Quanti-Max™ WST-8 Cell Viability Assay Kit (Biomax, Gangseo-gu, Korea). HepG2 cells (2500 cells/well) were seeded in 96-well plate (Corning, VA, USA) and incubated at 37 °C in a 5% CO_2_ humidified incubator. Forty-eight hours after glucose and/or PA treatment, the medium was removed and replaced with normal medium containing WST-8. The amount of WST-8 should be 10% of the volume of the medium. The cells were incubated in an incubator for 2 h. Absorbance was measured at 450 nm by using a microplate reader (Biochrom, Cambourne, UK).

### 2.5. Oil Red O Staining

After the treatment of HepG2 cells for 48 h, cells were washed with PBS and fixed with 10% formalin (BIOSESANG, Gyeonggi-do, Korea). Cells were stained with Oil Red O staining solution for 15 min, at room temperature, in the dark, after an addition of iso-propanol (Junsei Chemical Co. Ltd., Tokyo, Japan), for 5 min. The stained cells were photographed by using phase-contrast microscope (Motic, Xiamen, China).

### 2.6. Quantitative PCR

Total RNA was isolated from HepG2, using TRIzol (Life Technologies Inc, Carlsbad, CA, USA) in accordance with the manufacturer’s guideline. The concentration of total RNA was determined by spectrophotometer (Nanofroplite, Thermo Fisher Scientific Inc, Waltham, MA, USA). In accordance with the manufacturer’s guidelines, complementary DNA (cDNA) was synthesized by RNA reverse transcription, using RevertAid First Strand cDNA Synthesis Kit (Thermo Fisher Scientific, Waltham, MA, USA). Maxima SYBR Green/ROX qPCR Master Mix (2X) (Thermo Scientific, Waltham, MA, USA) was used for quantitative PCR (qPCR) according to the manufacturer’s guideline. Then qPCR was conducted in duplicate with SimpliAmp Thermal Cycler (Thermo Fisher Scientific, Waltham, MA, USA) and QuantStudio 3 (Thermo Fisher Scientific, Waltham, MA, USA). The quantity of mRNAs was normalized into the quantity of β-actin; an internal control was calculated by using the ΔΔCt value. All data were expressed as a relative quantity to each control value. The primer sequences of qPCR used in the experiment are shown in Table 1.

### 2.7. Statistical Analysis

All data were represented by the mean with standard error of the mean. The results were statistically analyzed by using a paired *t*-test or one-way analysis of variance (ANOVA), and Duncan post hoc analysis used SAS 9.4 (SAS Inc., Cary, NC, USA). Moreover, the mean difference between the control and amino acids pretreatment in the same treatment group was analyzed by *t*-test. The results were considered statistically significant at *p* < 0.05.

## 3. Results

### 3.1. Cell Viability

To examine the cytotoxicity of glucose and PA, and the effects of amino acid supplementation on cell proliferation, cell viability was assessed by WST-8 Cell Viability Assay Kit. As shown in Figure 1, treatment with glucose and/or palmitic acid resulted in a significant decrease in HepG2 cell survival. The betaine (Figure 1b) and taurine (Figure 1c) pretreatment significantly enhanced cell viability whether in glucose and/or PA treatment groups. The pretreatment of SAMe only increased cell viability in the glucose-treated group, and it seems to play a role in preventing glucose toxicity (Figure 1a).

### 3.2. Effects of Amino Acids on TG Accumulation in Glucose- and/or PA-Treated HepG2 Cells

In the PA and glucose with PA groups, treatment with 0.5 mM PA led to a dramatic increase in the number of lipid droplets in HepG2 cells compared with that in the normal control group (Figure 2). The results of Oil Red O staining showed that, after pretreatment with SAMe, betaine, and taurine, intracellular lipid droplets decreased in the PA-treated groups.

### 3.3. Glucose Metabolism after Amino Acids Pretreatment with Glucose and/or PA Treatment

To confirm the uptake of glucose after glucose treatment, the mRNA expression of GLUT2, a membrane-bound glucose transporter, was quantified by qPCR. After glucose treatment for 48 h, the GLUT2 mRNA expression generally increased in all groups (Figure 3a). In addition, GLUT2 expression is significantly inhibited by SAMe pretreatment (Figure 3a) (*p* < 0.05). Betaine and taurine had no significant effect on GLUT2 mRNA expression with glucose treatment (Figure 3a). Once glucose is absorbed by liver cells, it is phosphorylated by GK to glucose-6-phosphate. In order to know the regulation of amino acids on glucose metabolism, the mRNA expression of GK, a rate-limiting enzyme in glycolysis, was measured by qPCR. SAMe treatment significantly downregulated GK mRNA expression in the control group and the glucose-treatment group (Figure 3b). Meanwhile, in the group treated with glucose and PA, SAMe and BCAA upregulated GK expression (*p* < 0.05). Betaine (Figure 3b) and taurine (Figure 3b) had no significant effect on GK mRNA expression. Glucose-6-phosphatase (G6Pc) participates in the last step of gluconeogenesis. G6Pc mRNA expression was measured by qPCR to determine the regulation of amino acids on gluconeogenesis in a high-glucose environment. As shown in Figure 3c, SAMe significantly reduced the expression of G6Pc in glucose- and/or PA-treatment groups (*p* < 0.05). The other amino acids had no significant effect on G6Pc expression. The mRNA expression level of PCK1, a rate-limiting enzyme in gluconeogenesis, was quantified by qPCR to determine the regulation of amino acids on gluconeogenesis in a high-glucose environment. SAMe pretreatment decreased the PCK1 mRNA expression level in the glucose and the glucose and PA groups (Figure 3d). However, in the glucose-and-PA-treatment group, the pretreatment of BCAA stimulated the expression of PCK1 (Figure 3d) (*p* < 0.05). The other amino acids had no significant effect on PCK1 expression (Figure 3d).

### 3.4. Genes Involved in Lipid Metabolism after Amino Acids Pretreatment and Glucose and/or PA Treatment

Acetyl-CoA carboxylase 2 catalyzes the carboxylation of acetyl-CoA to malonyl-CoA. In order to determine the effect of amino acids on lipogenesis, the ACC2 mRNA expression level was measured by qPCR. As shown in Figure 4a, SAMe pretreatment reduced ACC2 expression in the glucose-treatment group (*p* < 0.05). Other amino acids had no significant effect on ACC2 expression (Figure 4a). In order to determine the effect of amino acids on lipogenesis, FASN mRNA expression level was measured by qPCR. As shown in Figure 4b, the mRNA expression of FASN was significantly increased by glucose treatment. Pretreatment with SAMe, taurine, and BCAA (Figure 4b) reduced FASN mRNA expression. Moreover, SAMe also downregulated FASN expression in the PA group (Figure 4b) (*p* < 0.05). Betaine had no significant effect on FASN expression (Figure 4b). Stearoyl-CoA desaturase (SCD1) plays a central role in lipogenesis and TG synthesis, and the mRNA expression of SCD1 was detected by qPCR. As shown in Figure 4c, the SCD1 expression level increased significantly in PA, glucose, and glucose and PA treatment groups compared to the control group (*p* < 0.05). SAMe pretreatment significantly reduced SCD1 expression in glucose, glucose, and PA treatment group (Figure 4c) (*p* < 0.05).

### 3.5. Regulation of Glucose and Lipid Metabolism after Amino Acids Pretreatment and Glucose and/or PA Treatment

Sterol regulatory element-binding protein-1c (SREBP-1c) is a key regulator of liver lipid metabolism and regulates ACC2 and FASN transcription, together with ChREBP. The mRNA expression of SREBP-1c was detected by qPCR to determine whether amino acids affect the expression of metabolism-related genes by regulating SREBP-1c. As shown in Figure 5b, SREBP-1c mRNA expression was increased after glucose and PA treatment. SAMe pretreatment inhibited SREBP-1c expression in the glucose-treated group (Figure 5b). Tau pretreatment also reduced SREBP-1c expression in the control and PA treatment group (Figure 5b) (*p* < 0.05). Other amino acids had no significant effect on its expression (Figure 5b). Carbohydrate-response element-binding protein (ChREBP) can regulate the transcription of SCD 1. Therefore, the expression of ChREBP was determined by qPCR. As shown in Figure 5c, the expression of ChREBP was increased significantly after glucose-, PA-, and glucose-with-PA-treatment groups. SAMe pretreatment inhibited the expression of ChREBP (Figure 5c) (*p* < 0.05). The effect of other amino acids on ChREBP mRNA expression was not obvious (Figure 5c). As a transcription factor, PPAR-γ can regulate the expression of ACC, FASN, and GK. The PPAR-γ mRNA expression level was significantly upregulated after glucose and PA treatment (Figure 5a), with BCAA pretreatment toward a further increase in the expression (Figure 5a). Meanwhile, the pretreatment of SAMe (Figure 5a) significantly downregulated the expression of PPAR-γ (*p* < 0.05). The effect of betaine and taurine on PPAR-γ mRNA expression was not obvious (Figure 5a).

## 4. Discussion

Typically, nonalcoholic fatty liver disease and type 2 diabetes are the symptoms of metabolic syndrome. NAFLD and type 2 diabetes are often linked to abnormal glucose and lipid metabolism. The liver is one of the major tissues for energy metabolism. Therefore, improving liver functions for energy metabolism is of great significance in regard tto metabolic syndrome. HepG2 cells were widely used to simulate liver function in vitro. Hence, in this study, we treated HepG2 with a high concentration of glucose (50 mM) and PA (0.5 mM) to simulate a high-glucose and high-fat environment. Some studies show that SAMe deficiency promoted the development of NAFLD in MAT1A knocked-out mice, and supplementation with betaine and taurine to high-fat-diet-induced obese mice improved hepatic steatosis and restored insulin sensitivity [9,10,21]. BCAA is one of the widely studied amino acids, and recent studies have shown that BCAAs have a positive effect on glucose metabolism, but an increased risk of insulin resistance [22,23]. Therefore, we focused on SCAA and BCAA, the two group of amino acids that have an impact on NALFD and insulin resistance. Since the mechanism of the effects of SCAA and BCAA on NALFD and insulin resistance is still unclear, this study was designed to evaluate the role of SCAA and BCAA in glucose and lipid metabolism.

In this study, we evaluated the effects of SCAA and BCAA on the cell viability of HepG2 cells induced by glucose and/or PA at first. High glucose was reported to induce apoptosis in different cell lines, and PA was also found to induce apoptosis in various hepatocytes [24,25,26,27,28]. Similar to the results of these studies, when glucose and PA, in our experiment, were treated separately, the cell viability of HepG2 decreased slightly, and HepG2’s viability decreased significantly when glucose and PA were treated together (Figure 1). In this study, S-adenosylmethionine, as a precursor to GSH, may also inhibit cell death by reducing ROS levels. Therefore, the results show that SAMe could only improve cell viability in the glucose-treated group; taurine and betaine could effectively improve cell activity in the glucose- and/or PA-treatment group, possibly through against oxidative stress.

Palmitic acid is known to induce steatosis of hepatocytes and increase the dose-dependent content of TG in hepatocytes. Some studies have shown that, in mice fed a high-fat diet, taurine supplementation significantly reduced liver triglyceride levels, and betaine supplementation significantly increased liver SAMe concentration and reduced liver TG accumulation [9]. The supplementation of BCAA to the diets of obese mice also reduced liver weight and triglyceride content in high-fat-diet-induced obese mice [18]. After pretreatment with SAM, betaine, taurine, and BCAA intracellular lipid droplets decreased in the PA-treated groups. However, all amino acids had no significant effect on TG accumulation in glucose and glucose with the PA-treated group. Therefore, the above amino acids only inhibit the TG accumulation caused by PA treatment.

To investigate whether amino acids affect fat accumulation by regulating enzymes related to glucose and fat metabolism, we determined if amino acids improved glucose metabolism of glucose- and/or PA-treated HepG2 cells. The liver regulates glucose homeostasis mainly by maintaining the balance between glycogenesis and glycolysis [29]. We first observed the mRNA expression levels of the key glucose transporter, GLUT2, which is the main transporter of glucose uptake across the hepatocyte plasma membrane. In the state of glucose metabolism disorder, GLUT2 can increase the level of hepatic glucose output in parallel with enhancement of gluconeogenesis [30]. In the liver, once glucose metabolism disorder arises in a high-glucose and/or -fat environment, the level of GLUT2 expression would increase [31]. We observed the reduction of GLUT2 by SAMe in a high-glucose environment, which confirmed the beneficial effect of SAMe on hepatic glucose homeostasis (Figure 3a). In several studies, increased GK activity was found in obese diabetic patients and obese Zucker rats with hyperinsulinemia [32,33]. The results of this study showed that the expression of GK in the glucose with PA treatment group was increased, and SAMe downregulated the expression of GK in general (Figure 3b). The factors influencing GK need further study. Some studies have shown that, in animal experiments, overexpression of PCK1 promoted insulin resistance [34]. A high glucose concentration caused transcriptional activation of G6Pc, and overexpression of G6Pc can lead to increased endogenous glucose production in the liver [35,36]. Furthermore, the results of this research suggested that SAMe inhibited glucose synthesis by significantly reducing the level of gluconeogenic enzyme G6Pc (Figure 3c) and PCK1 (Figure 3d) in HepG2 cells (*p* < 0.05). In addition, BCAA increased the expression of GK and PCK1 significantly (*p* < 0.05) in the high-glucose and high-fat environment, and this may promote the synthesis of glucose. However, betaine and taurine had no significant effect on GLUT2, GK, G6Pc, and PCK1 mRNA expression.

We next evaluated the effects of amino acids on lipid metabolism of HepG2 cells in high-glucose and/or -fat environments. The pretreatment of SAMe significantly inhibited the mRNA expression of ACC2 (Figure 4a), FASN (Figure 4b), and SCD1 (Figure 4c) (*p* < 0.05) to different degrees, especially in the high-glucose environment. Taurine and BCAA pretreatment also reduced FASN expression. The attenuated mRNA expression level of FASN indicated the repression of de novo lipogenesis. In HepG2 cells, SAMe inhibited lipid synthesis and lipid accumulation by regulating ACC2, FASN, and SCD1.

The gene expressions of these lipogenic enzymes are known to be mainly regulated by two transcriptional factors, SREBP-1c and ChREBP [37,38]. Overexpression of SREBP-1c was found in obese animal models and in patients with hepatic steatosis, suggesting that this transcription factor is involved in the pathogenesis of obesity-related diseases [39,40]. In Figure 5b, we could see that, in the high-glucose and/or high-fat environment, the expression of SREBP-1c increased, and SAMe pretreatment prevented the upregulated mRNA expression of SREBP-1c. In this study, SAMe pretreatment significantly reduced the mRNA expression of ChREBP in the glucose treatment group (Figure 5c) (*p* < 0.05). The expression of lipogenic transcription factors SREBP-1c and ChREBP were reduced, thus providing a possible mechanism for the ACC2, FASN, and SCD 1 inhibition as an upstream transcriptional regulation of lipogenesis. Peroxisome-proliferator-activated receptor gamma (PPAR-γ) is a ligand-activated transcription factor that belongs to the nuclear receptor superfamily and controls the expression of genes involved in organogenesis, inflammation, cell differentiation, proliferation, and lipid and carbohydrate metabolism [41]. Many studies have shown that excessive PPAR-γ promotes hepatic steatosis [42,43]. Our study demonstrated that PPAR-γ expression increased in PA-treated HepG2 cells, while SAMe pretreatment effectively inhibited the expression of PPAR-γ (Figure 5a) (*p* < 0.05). However, in the high-glucose and high-fat environment, the pretreatment of BCAA further improved the expression of PPAR-γ. The results above showed that SAMe had significant effects on the glucose and lipid metabolic process, and the downregulation of PPAR-γ was significant. PPAR-γ is highly expressed in adipocytes and plays a role in glucose homeostasis and adipocyte differentiation [44]. Although the expression of PPAR-γ in liver is relatively low, multiple studies have shown that it can selectively upregulate the content of fat-generating enzymes in liver cells, thus promoting lipid synthesis and fat accumulation [42,43]. On the other hand, the liver-specific deletion of PPAR-γ in db/db mice significantly improved hepatic steatosis, showing decreased expression of ACC, FASN, and SCD1 [45]. These results may be due to the fact that activation of PPAR-γ can enhance the transcription factor SREBP-1c. However, it is still unclear whether PPAR-γ directly or indirectly activates SREBP-1c. In addition, Yu et al. reported that, when PPAR-γ is overexpressed in the liver, there was a significant increase in the expression levels of PCK1 and GLUT2, which promote fat accumulation [42]. This is consistent with our experimental results, thus making it possible to presume the underlying mechanisms used when SAMe pretreatment decreased the expression of PPAR-γ, followed by the downregulation of mRNA expressions of GLUT2, PCK1, ACC2, FASN, and SCD1.

Recent studies have shown that SREBP-1c is the main mediator of insulin acting on GK and fat gene expression in the liver, and SREBP-1c is directly involved in the expression of GK gene [46]. Consistent with the above, in our experiment, SAMe decreased the mRNA expression of GK in the glucose environment. Moreover, SAMe increased the GK mRNA expression in glucose with the PA treatment group, possibly by increasing glucose utilization and synthesizing glycogen. It is hard to explain the SAMe regulation of mRNA GK expression. However, in this study, we only confirmed the expression changes of major genes regulating glucose metabolism following SAMe treatment. In future studies, it should be further elucidated how the expression of the corresponding genes is regulated.

We have known that ChREBP and SREBP synergistically induce the expression of the lipogenic gene. Accumulated studies have shown that, in ChREBP knockout ob/ob mice, the symptoms of metabolic syndrome were improved, such as obesity, insulin resistance, and fatty liver. A recent study has shown that, in liver-specific ChREBP knockout mice, fatty liver and glucose intolerance were improved, while liver-specific SREBP-1c and PPAR-γ knockout mice could only improve fatty liver. G6Pc is a key enzyme that regulates gluconeogenesis and glucose output of the liver, and decreased G6Pc activity was found in liver-specific ChREBP inhibited mice [40]. Moreover, ChREBP may regulate the expression of the G6Pc gene at the transcriptional level. The results were consistent with our study: SAMe pretreatment inhibited the expression of ChREBP, the G6Pc expression was also decreased, and gluconeogenesis was inhibited.

In summary, SAMe has a negative regulatory effect on the three transcription factors, PPAR-γ, SREBP-1c, and ChREBP, in a high glucose environment. The expression of PPAR-γ was downregulated in high-glucose and/or high-fat environments. SAMe mainly affects glucose and fat-metabolism-related genes GLUT2, PCK1, ACC2, FASN, and SCD1 by regulating PPAR-γ and inhibits fat accumulation. Although the regulation of glucose and lipid metabolism by PPAR has been studied in the previous studies, the effect of SAMe on the expression of PPAR and its regulatory mechanism remains to be elucidated for further studies. Although the pretreatment of taurine and betaine significantly increased the viability of cells and inhibited the fat accumulation, it had no significant effect on the expression of enzyme correlated with glucose and fat metabolism. Different from our hypothesis, BCAA increased PPAR-γ and PCK1 expression in the high-glucose and high-fat environment and decreased the expression of FASN; this seems to confirm that BCAA has a positive effect on obesity, but an increased risk of T2DM.

## 5. Conclusions

The effect of sulfur-containing amino acids (SCAAs) on metabolic syndrome appears to be positive, particularly for SAMe. The supplementation of SAMe can regulate the genes related to fat and glucose metabolism by inhibiting the expression of transcription factor PPAR-γ, followed by modulating the downstream genes, so as to change the intracellular lipid metabolism, to regulate glucose homeostasis, and to inhibit the accumulation of intracellular triglycerides, thus possibly preventing metabolic syndrome. In addition, the supplementation of BCAA inhibited the lipogenesis-related enzyme (FASN) mRNA expression but increased the mRNA expression of the gluconeogenesis-related enzyme (PCK1).

## Figures and Tables

**Figure 1 nutrients-14-03050-f001:**
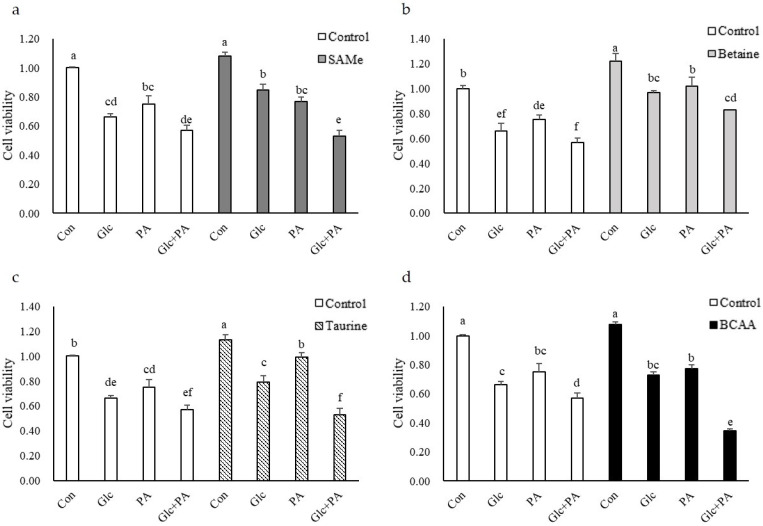
Effects of (**a**) SAMe, (**b**) betaine, (**c**) taurine, and (**d**) BCAA pretreatment and glucose and PA treatment on cell viability of HepG2 cells. Different alphabets indicate significant differences (*p* < 0.05). The graphs represent means, and error bars indicate the standard error of the mean (SEM) (n = 3).

**Figure 2 nutrients-14-03050-f002:**
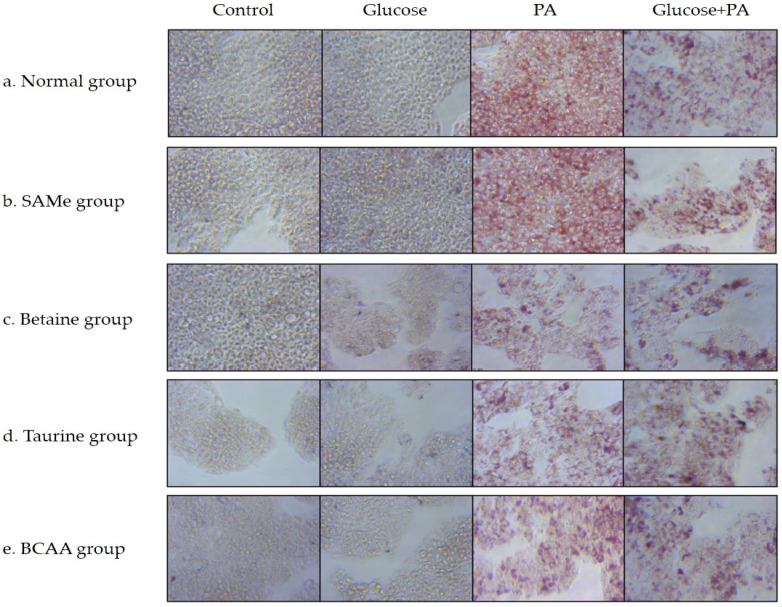
Effects of amino acids on TG accumulation in glucose- and/or PA-treated HepG2 cells.

**Figure 3 nutrients-14-03050-f003:**
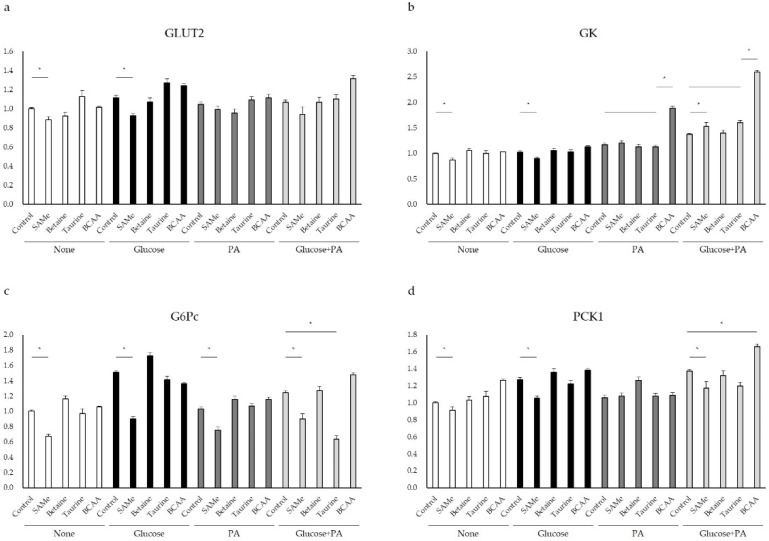
Relative mRNA expression levels of genes related to glucose metabolism: (**a**) GLUT2, (**b**) GK, (**c**) G6Pc, and (**d**) PCK1. Asterisk indicate significant differences (*p* < 0.05). The graphs represent means, and error bars indicate the standard error of the mean (SEM) (n = 3).

**Figure 4 nutrients-14-03050-f004:**
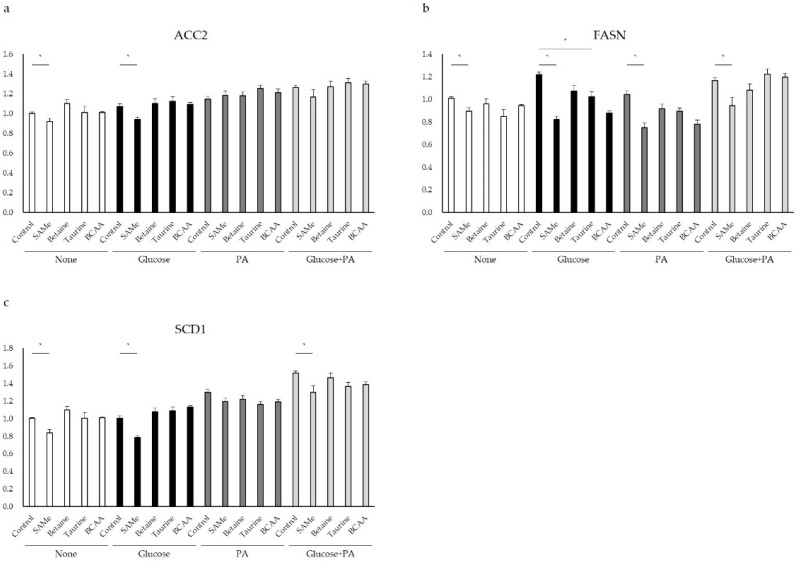
Relative mRNA expression levels of genes related in lipid metabolism: (**a**) ACC2, (**b**) FASN, and (**c**) SCD1. Asterisk indicate significant differences (*p* < 0.05). The graphs represent means, and error bars indicate the standard error of the mean (SEM) (n = 3).

**Figure 5 nutrients-14-03050-f005:**
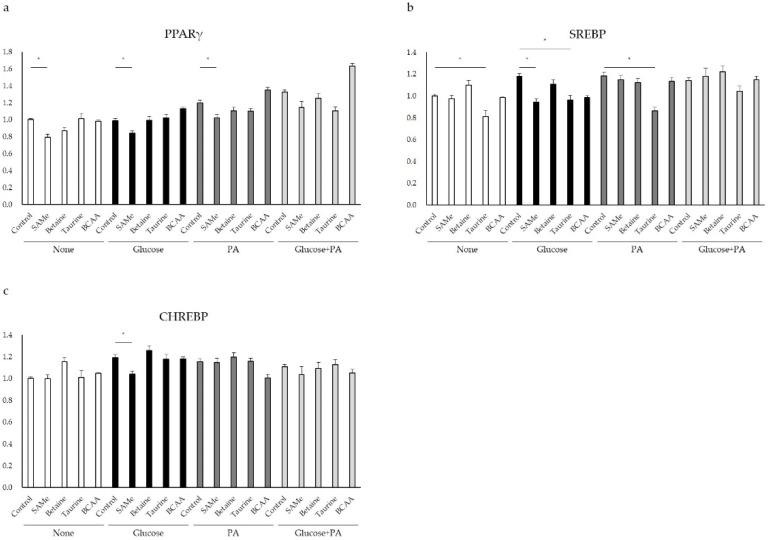
Relative mRNA expression levels of genes related in lipid metabolism: (**a**) PPARγ, (**b**) SREBP, and (**c**) CHREBP. Asterisk indicate significant differences (*p* < 0.05). The graphs represent means, and error bars indicate the standard error of the mean (SEM) (n = 3).

**Table 1 nutrients-14-03050-t001:** Primer sequences.

Gene	mRNA No.	Primer Sequence	
ACTB	NM_001101.5	Forward	GGACTTCGAGCAAGAGATGG
		Reverse	AGGAAGGAAGGCTGGAAGAG
GLUT2	NM_000340.2	Forward	GCGACGTTCTCTCTTTCTAAT
		Reverse	GGCTATCATGCTCACATAACT
GK	NM_000167.5	Forward	GAAACTACACTGTCCCATCTC
		Reverse	GGATTGAGGTGTTGCCTATC
PCK1	NM_002591.4	Forward	GCCTGGATGAAGTTTGACGC
		Reverse	ATGGCATTGGGGTTGGTCTT
G6PC	NM_000151.4	Forward	GCAATGGGCACTGGTATT
		Reverse	GGAGTCACACATGGGAATAAG
ACC2	NM_001093.4	Forward	GGAACATCCCTACGCTAAAC
		Reverse	GACAAGGTGGAGTGAATGAG
SCD1	NM_005063.5	Forward	CAACTACCACCACTCCTTTC
		Reverse	GAGACTTTCTTCCGGTCATAG
FASN	NM_004104.5	Forward	GGTTTGATGCCTCCTTCTT
		Reverse	GGAGTGAATCTGGGTTGATG
SREBP-1	NM_001018067.2	Forward	TACCGCTCCTCCATCAAT
		Reverse	GTGTTGCAGAAAGCGAATG
ChREBP	NM_032951.3	Forward	GACAGCTGAGTACATCCTTATG
		Reverse	TGCTGGCACAGGTTAATG
PPARγ	NM_001330615.1	Forward	GTCGTGTCTGTGGAGATAAAG
		Reverse	GGATCCGACAGTTAAGATCAC

## Data Availability

Not applicable.

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
