# Peer review of "Effects of Amino Acids Supplementation on Lipid and Glucose Metabolism in HepG2 Cells"

_nutrients, 2022, doi:10.3390/nu14153050_

Round 1
Reviewer 1 Report
It can be said that the title of the article adequately reflects the subject of the article.
I think the abstract summarizes the article adequately.
Introduction: The purpose of the article can be stated more clearly.
Material and Methods: I think the methods used are appropriate. The tests used for statistical analysis are suitable. The t-test used should be specified (such as paired or unpaired t-test).
Tables and figures are sufficient.
I think that the findings are discussed sufficiently with reference.
The references were selected between 1983-2019; can be considered current.
Author Response
Dear Reviewer 1
Thank you very much for your valuable comments for this manuscripts.
We are trying to restate and correct the some parts of the manuscripts based on your comments as below
Introduction: The purpose of the article can be stated more clearly.
A: The purpose of the study has been restated more clearly
Material and Methods: I think the methods used are appropriate. The tests used for statistical analysis are suitable. The t-test used should be specified (such as paired or unpaired t-test).
A: We specified the specific method for statistics for the results.
The references were selected between 1983-2019; can be considered current.
A: The reference prior to the year 2000 was replaced by more recent reference.
All the changes were highlighted in red for your convenience to review.
Thank you

Reviewer 2 Report
Dear authors
thank you for this paper, well written and investigating an interesting mechanism.
can i please ask you if you can make these minor changes:
1. add some key numerical data of statistical analysis on your abstract.
2. expand upon limitations of this study.
thank you
Author Response
Dear Reviewer 2
Thank you very much for your valuable comments for this manuscripts.
We are trying to restate and correct the some parts of the manuscripts based on your comments as below
1. add some key numerical data of statistical analysis on your abstract.
A: We added some numerical data in the abstract part for the clearer summary.
2. expand upon limitations of this study.
A: We added more limitation part for the research as requested.
All the changes were highlighted in red for your convenience to review.
Thank you
